# Diversity and distribution of sediment bacteria across an ecological and trophic gradient

Hailey M. Sauer[1,2], Trinity L. Hamilton[1,3]*, Rika E. Anderson[4], Charles E. Umbanhowar, Jr.[5], Adam J. Heathcote[2]

**1** Department of Plant and Microbial Biology, University of Minnesota, St. Paul, Minnesota, United States of America, **2** St. Croix Watershed Research Station, Science Museum of Minnesota, Marine on St. Croix, Minnesota, United States of America, **3** The Biotechnology Institute, University of Minnesota, St. Paul, Minnesota, United States of America, **4** Biology Department, Carleton College, Northfield, Minnesota, United States of America, **5** Department of Biology and Environmental Studies, St. Olaf College, Northfield, Minnesota, United States of America

* trinityh@umn.edu

**Data Availability Statement:** All relevant data are within the paper and its Supporting information

## Abstract

The microbial communities of lake sediments have the potential to serve as valuable bioindicators and integrators of watershed land-use and water quality; however, the relative sensitivity of these communities to physio-chemical and geographical parameters must be demonstrated at taxonomic resolutions that are feasible by current sequencing and bioinformatic approaches. The geologically diverse and lake-rich state of Minnesota (USA) is uniquely situated to address this potential because of its variability in ecological region, lake type, and watershed land-use. In this study, we selected twenty lakes with varying physio-chemical properties across four ecological regions of Minnesota. Our objectives were to (i) evaluate the diversity and composition of the bacterial community at the sediment-water interface and (ii) determine how lake location and watershed land-use impact aqueous chemistry and influence bacterial community structure. Our 16S rRNA amplicon data from lake sediment cores, at two depth intervals, data indicate that sediment communities are more likely to cluster by ecological region rather than any individual lake properties (*e.g.*, trophic status, total phosphorous concentration, lake depth). However, composition is tied to a given lake, wherein samples from the same core were more alike than samples collected at similar depths across lakes. Our results illustrate the diversity within lake sediment microbial communities and provide insight into relationships between taxonomy, physicochemical, and geographic properties of north temperate lakes.

## Introduction

A community of microorganisms living together in a particular environment or habitat are referred to as a microbiome. In the past decade, studies into the microbiomes of human organs, plants, soils, waters, and even space station astronauts have enhanced our

files. All 16S rRNA amplicon data are available from the SRA database at BioProject PRJNA763898.

**Funding:** Funding for this project was provided by a grant to AJH from the Minnesota Environment and Natural Resources Trust fund as recommended by Legislative-Citizen Commission on Minnesota Resources (LCCMR). https://www.lccmr.leg.mn/ HMS and TLH were also supported by NSF grant #1948058. The funders had no role in study design, data collection and analysis, decision to publish, or preparation of the manuscript.

**Competing interests:** The authors have declared that no competing interests exist.

understanding of how microbiota protect us from pathogens, increase agricultural production, and ultimately cycle nutrients in and throughout the natural environment [1–4]. Microbiomes are connected to the biogeochemical cycling of nutrients at both local and global levels, and as a result, disturbances or variations in their community composition can result in gains or losses of functional attributes, changes in nutrient availability, and shifts in ecosystem adaptability [5–7]. Understanding the selective pressures on microbiome community composition can provide insight into an environment's ability to support higher trophic levels and respond to anthropogenic change. Freshwater lakes are an ideal system to explore these insights as they provide a variety of regulating and cultural services, which hinge on the composition of their microbiome [8].

Despite their relatively small surface area, lakes contribute disproportionately to biogeochemical cycles, including the essential macronutrients carbon, nitrogen, and phosphorus [9–13]. Nutrients in lakes are recycled through several biotic and abiotic processes, but ultimately a significant proportion end up in sediments, where bacterial abundance and diversity typically exceeds that of the water column [14–16]. In the sediments, bacteria and archaea degrade organic matter—consuming oxygen and proceeding with anaerobic respiration processes. These respiratory processes occur along a redox gradient, eventually leading to the transformation of nitrogen, iron, and sulfur compounds. The complementary metabolisms of sediment microbiomes make sediments a global biogeochemical hotspot, one in which there has been a concerted effort to understand the environmental factors that regulate composition and function [17–20].

Chemical and physical characteristics of the lake such as salinity, pH, temperature, and nutrient concentrations select for specific bacteria, a process commonly referred to as species sorting [21]. The physicochemical characteristics of the system are partly based on the external inputs of both organic matter and nutrients, such as phosphorus and nitrogen, from the surrounding watershed [22]. Different land uses (e.g., agricultural, urban, forested, etc.) in the watershed strongly influence the amount and types of terrestrial organic matter and nutrients that enter the water, and therefore land use may subsequently affect community composition [23]. While there have been several studies that address the effects that local environmental factors have on microbiome species selection (e.g., eutrophic reservoirs, alpine lakes), few have examined the effects of the land-use of the watershed on bacterial community assembly [22–30].

In this study we selected twenty lakes with varying physio-chemical properties across four ecological regions with varying land use in Minnesota (U.S.A.) (Fig 1). We sought to (i) evaluate the diversity and spatial variation of the bacterial community at the sediment-water interface and (ii) determine how lake location and watershed land-use impact aqueous chemistry and influence bacterial community structure. We hypothesize that community composition of lake sediments will appear homogeneous across ecological regions and land use at higher taxonomic levels; however, we hypothesize increased structure by eco-region at lower taxonomic levels. To test this, we compare the alpha and beta diversity of bacteria across taxonomic scales (Phylum to Order) and ecological regions, and we highlight important regional and local factors that influence community composition.

## Materials & methods

### Site description

For this study, we selected twenty lakes within Minnesota's Sentinel Lakes in a Changing Environment (SLICE) program. SLICE is a collaborative research initiative providing long-term data on a representative sub-sampling of Minnesota's lakes that span the diverse geographic,

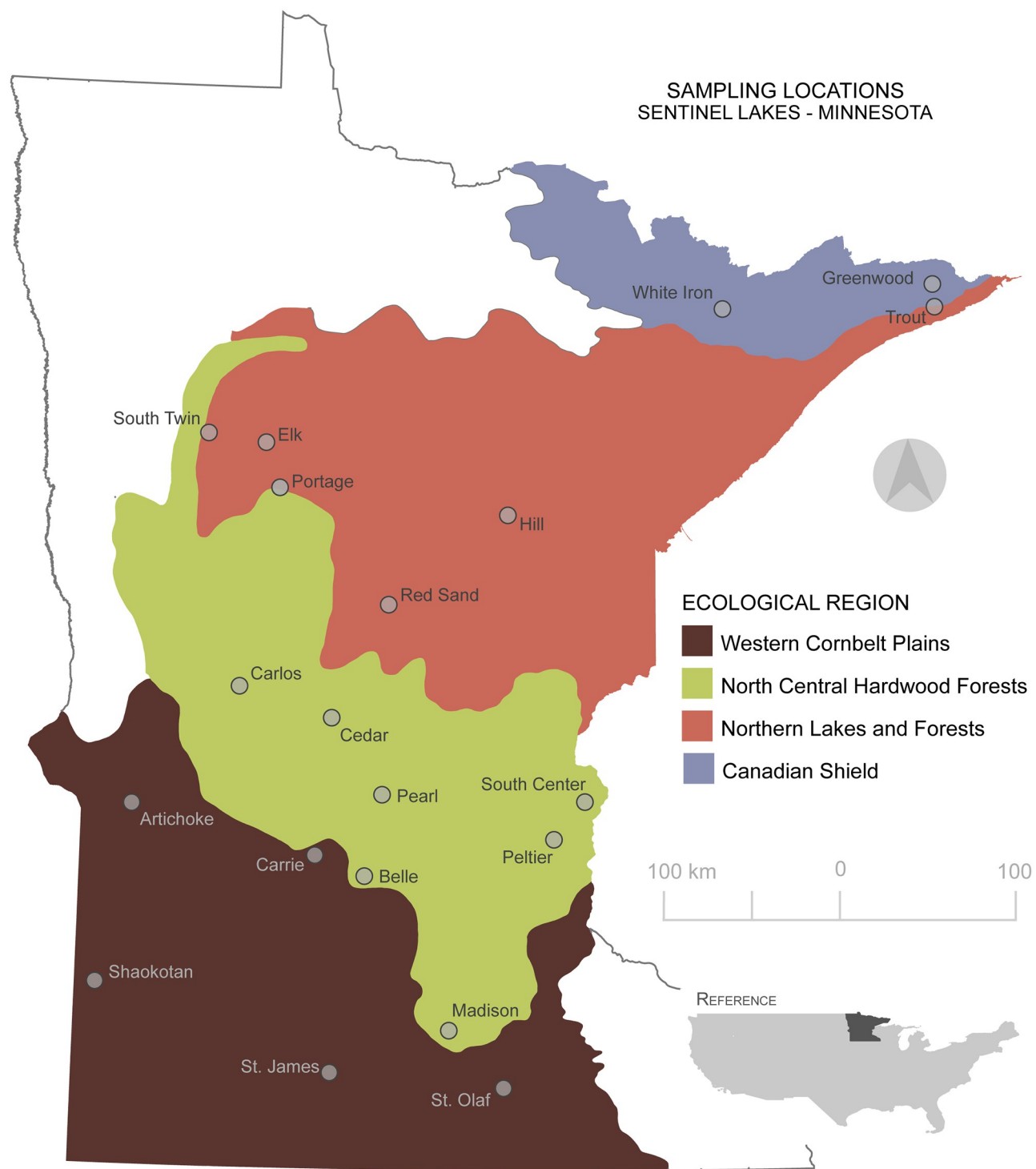

**Fig 1. Map of sampling locations.** Location of the twenty study lakes that were cored across the state of Minnesota (U.S.A.) between the summer of 2018 and 2019, shaded by ecoregion. Map was created using QGIS and data were made available by the MN Geospatial Commons (public domain) https://gisdata.mn.gov/.

land-use, and climatic gradients present in Minnesota (Fig 1). The lakes span four of the seven Environmental Protection Agency/Commission for Environmental Cooperation's (level III) ecological regions. These regions can be characterized by their underlying geology, soils, vegetation, and land use (S1 Table). This is the first comprehensive sediment bacterial survey of these lakes.

## Water sample collection & analysis

From each site we collected water profile measures for temperature, pH, conductivity, turbidity, and dissolved oxygen using a YSI XO2 multi-parameter sonde (YSI, Inc.). We also collected an integrated (0-2m) epilimnetic water sample, and a hypolimnetic (maximum lake depth–1m) water sample when thermal stratification was present. All samples were stored on ice in the field and at 4°C or -20°C in the laboratory, depending on methodology, until processed. Samples for soluble reactive phosphorus (SRP), dissolved organic carbon (DOC), and dissolved inorganic carbon (DIC) were filtered, processed, and analyzed within 36 hours of sampling using standard methods for SRP (4500-P) on a SmartChem 170 (Unity Scientific, Inc.) and DIC/DOC Method 5310-C using a Torch Combustion TOC Analyzer (Teledyne Tekmar, Inc.) [31]. Samples for total nitrogen (TN) and total phosphorus were frozen and analyzed using standard methods for TN (4500-N), and TP (4500-P). Samples for ammonia (NH$_3$) and nitrate (NO$_3$) were filtered and frozen prior to analysis following methods NH$_3$ (4500-NH$_3$) and NO$_3$ (4500-NO3). All TP, TN, NH3, NO3 samples were analyzed within six months of sampling on a SmartChem 170 (Unity Scientific, Inc.) discrete analyzer (APHA 2012). Additionally, we filtered, froze, and analyzed samples for chlorophyll-a concentrations via fluorometry following the EPA method 445.0 [32]. We provided a complete summary of aqueous chemistry results, including sampling dates, in the S2 Table.

## Sediment sample collection & DNA isolation

We collected sediment cores from July 2018 through June 2019 using a rod-driven piston corer with a 7cm diameter polycarbonate tube [33]. We determined coring locations (i.e., flat areas near the deepest basin) using publicly available bathymetric maps (https://www.dnr.state.mn.us/lakefind/index.html), avoiding steep-sided "holes" where sediment-focusing may be high. After sediment core retrieval, we stabilized core tops in the field using a gelling agent (e.g., Zorbitrol) and returned intact cores to the laboratory where we stored them vertically at 4°C for no more than seven days until processing. In cases where the upper sediments were extremely flocculent, we immediately sectioned the upper most sections (~0–30 cm) in the field to prevent mixing during transport.

   We vertically extruded the cores in the lab in 1 to 2 cm intervals, depending on lake productivity, and took subsamples from two intervals for DNA analysis. The subsamples collected were from the 0-2cm (hereafter referred to as shallow) and either the 3-4cm or 4-6cm interval (hereafter referred to as deep). Subsamples were frozen under nitrogen for up to three months before DNA was extracted (S3 Table). We extracted DNA from 0.25g of wet sediment from each subsample using a PowerSoil DNA Isolation Kit (Qiagen, Inc.) following the manufacturer's protocols. We performed negative controls by carrying out extractions on blanks, using only reagents without sample. We determined final bulk DNA concentrations using a Qubit™ dsDNA HS Assay kit (Molecular Probes, Eugene, OR, USA) and Qubit™ Fluorometer (Invitrogen, Carlsbad, CA, USA). The detection limit for the Qubit™ dsDNA HS Assay Kit is 10 pg/μL. All samples that yielded detectable amounts of DNA were sent for sequencing (S3 Table). Despite not detecting DNA in our negative controls, these were submitted for sequencing

where they failed to pass quality control performed by the University of Minnesota Genomic Center (UMGC) and no sequencing information was obtained.

## Nucleic acid preparation, amplification, and sequencing

We submitted the DNA samples to the UMGC where they performed library preparation for Illumina high-throughput sequencing using a Nextera XT workflow and a 2x300 bp chemistry. The workflow utilizes transposome-based shearing which fragments the DNA and adds adapter sequences in one step. The DNA was amplified and dual-indexed with adapter sequences through PCR, using primers 515F (5′-TCGTCGGCAGCGTCAGATGTGTATAAGA GACAGGTGCCAGCMGCCGCGGTAA-3′) and 806R (5′-GTCTCGTGGGCTCGGAGATGTGTA TAAGAGACAGGGACTACHVGGGTWTCTAAT-3′) to target the V4 hypervariable region of bacterial 16S SSU rRNA gene sequences. The amplicon library preparation methods created and employed by the UMGC have been shown to be more quantitatively accurate and qualitatively complete—detecting taxonomic groups that often go undetected with existing methods [34]. The indexed samples were then sequenced once using an Illumina MiSeq at the UMGC. A total of 3.29 million *(3,290,170)* raw reads were obtained from 40 samples.

## Data processing

We conducted post-sequence processing in Mothur (v1.43.0) following the MiSeq SOP [35, 36]. Briefly, we merged forward and reverse reads, and screened, trimmed and removed ambiguous bases. We then aligned the reads to references in the SILVA database (v.132), and identified and removed chimeras using vsearch (v2.13.3) [37, 38]. Finally, given the nature of the study (i.e., broad scale patterns of diversity), we classified the sequences as operational taxonomic units (OTUs) using a 97% similarity threshold and assigned taxonomy using the SILVA database [39, 40].

## Community analysis & statistics

Unless otherwise stated, we conducted all statistical analyses in R (v4.0.0) [41, 42]. We loaded both the environmental and community data into R using Phyloseq (v1.32.0) [43] and removed any reads classified as mitochondrial or chloroplast. Our final dataset after all post-processing contained 2,181,132 reads assigned to 53,854 taxa across 40 (two sediment depths/lake) samples.

**Alpha diversity.** We removed all singletons (OTUs observed only once across all 40 samples) from the data before calculating alpha diversity statistics. Given the observed correlation of richness based on sample read depth across sequencing batches (S1 Fig), we chose to rarefy the data to 90% the read depth of the lowest samples (15,771 reads; S2 Fig and S3 Table). Our final dataset for alpha diversity included 630,840 read counts of 25,563 taxa across 40 samples. We calculated alpha diversity measures using the Phyloseq package in R (S3 Fig and S4 Table) [43]. We compared the richness (observed number of OTUs) and evenness (Shannon) of the samples based on sample depth (shallow n = 20, deep n = 19) using a Wilcox test, and trophic status (hypereutrophic n = 4, eutrophic n = 16, mesotrophic n = 16, and oligotrophic n = 3) and ecological status (Western Cornbelt Plains n = 12, North Central Hardwood Forests n = 14, Northern Lakes & Forests n = 8, Canadian Shield n = 5) using a Kruskal-Wallis test with a Dunn Post Hoc test and Bonferroni correction. In all tests, one outlying sample (Trout, Deep) was removed due to uncharacteristically low diversity. Finally, we assessed the predictive capabilities of the environmental parameters, collected at the time of sampling (S2 Table), on the alpha diversity of the sample using multiple regression and determined the significance and variance partitioned by each regressor using the relaimpo (v.2.2.3) and vegan (v.2.5–6)

packages in R [44, 45]. We selected the final models based on AIC scores for both richness (observed) and evenness (Shannon).

**Beta diversity.** Prior to beta diversity analysis we filtered the samples by removing any OTU that did not have 2 or more counts and occur in at least 10% of the samples. Post filtering, the average number of reads per sample was reduced to 47,605, the minimum read depth was 15,150, and the maximum read depth was 99,561. Since OTU data have a strong positive skew, we attempted to diminish the effects using a variance stabilizing transformation (VST) [46]. Log-like transformations, like VST, have been shown to transform count data to near-normal distributions and produce larger eigengap values, ultimately leading to more consistent correlation estimates which influence downstream analyses [47]. After filtering and transformation, the final dataset for beta diversity analysis included 5,512 taxa across 40 samples.

We visualized the sample dissimilarity using principal component analysis (PCA) and the ordinate function in Phyloseq [43]. After ordination, we further analyzed the distribution of taxa based on the ecological regions using permutational analysis of variance (PERMANOVA) and the "adonis" function in vegan [45]. We used a Bray-Curtis dissimilarity to test for group differences and assessed dispersion within groups using permutations and vegan's "betadisp" and "permutest" functions. Prior to creating the dissimilarity matrix, we converted negative VST values to zero because negative values after transformation likely represent zero counts or very few counts and for the distances and hypothesis in future tests these values would be negligible. We performed a cluster analysis using Ward's (D2) method and the same dissimilarity matrix generated for the PERMANOVA analysis.

## Results and discussion

### Alpha diversity

We used alpha diversity metrics to summarize the structure of the bacterial communities in terms of the number of OTUs (richness) and the distribution of their abundances (evenness) for all samples. We then compared the observed diversity (a measure of richness) and Shannon diversity (a measure of evenness) across sampling locations and ecological regions (S4 Fig). Sample richness varied from ~2000–4000 OTUs and sample evenness varied from 5.5 to 7.5. Sediments, both shallow and deep, from Carrie Lake were the most diverse in terms of richness (4116 OTUs and 4100 OTUs; shallow and deep respectively) and evenness (7.41, 7.39; shallow and deep respectively). The least diverse shallow sample in terms of richness was Pearl Lake (2371 OTUs) and evenness was Greenwood Lake (6.44). Trout Lake was the least diverse deep sample in both the total number of OTUs (552 observed) and Shannon diversity (5.49). There was no significant difference within lake diversity between shallow and deep lake sediments across all samples (Wilcoxon test p >0.05); however, the deeper interval sample was more diverse in both richness and evenness in a majority of lakes. The exception to that pattern were the samples from lakes in the Canadian Shield (CS) where all of the shallow interval samples were more diverse.

While all samples were highly diverse when compared to the bacterial diversity of the overlying water column or the number diatom species found in the sediments (Observed Richness > 2250 OTUs; Shannon 5.5–7.5), there were differences in the levels of richness and evenness when comparing samples across the ecological region (Fig 2 and S5 Fig) [48–50]. Shannon diversity (evenness) levels were statistically different across the ecological regions (Kruskal Wallis p = 0.008). Samples from lakes in the Western Cornbelt Plains (CB) were more diverse than both the Northern Lakes and Forests (NLF) (Dunn's test p = 0.0206) and CS (Dunn's test p = 0.0099) samples. Observed diversity (richness) was also statistically significant across the ecological regions (Kruskal Wallis p = 0.003). Again, there were statistical

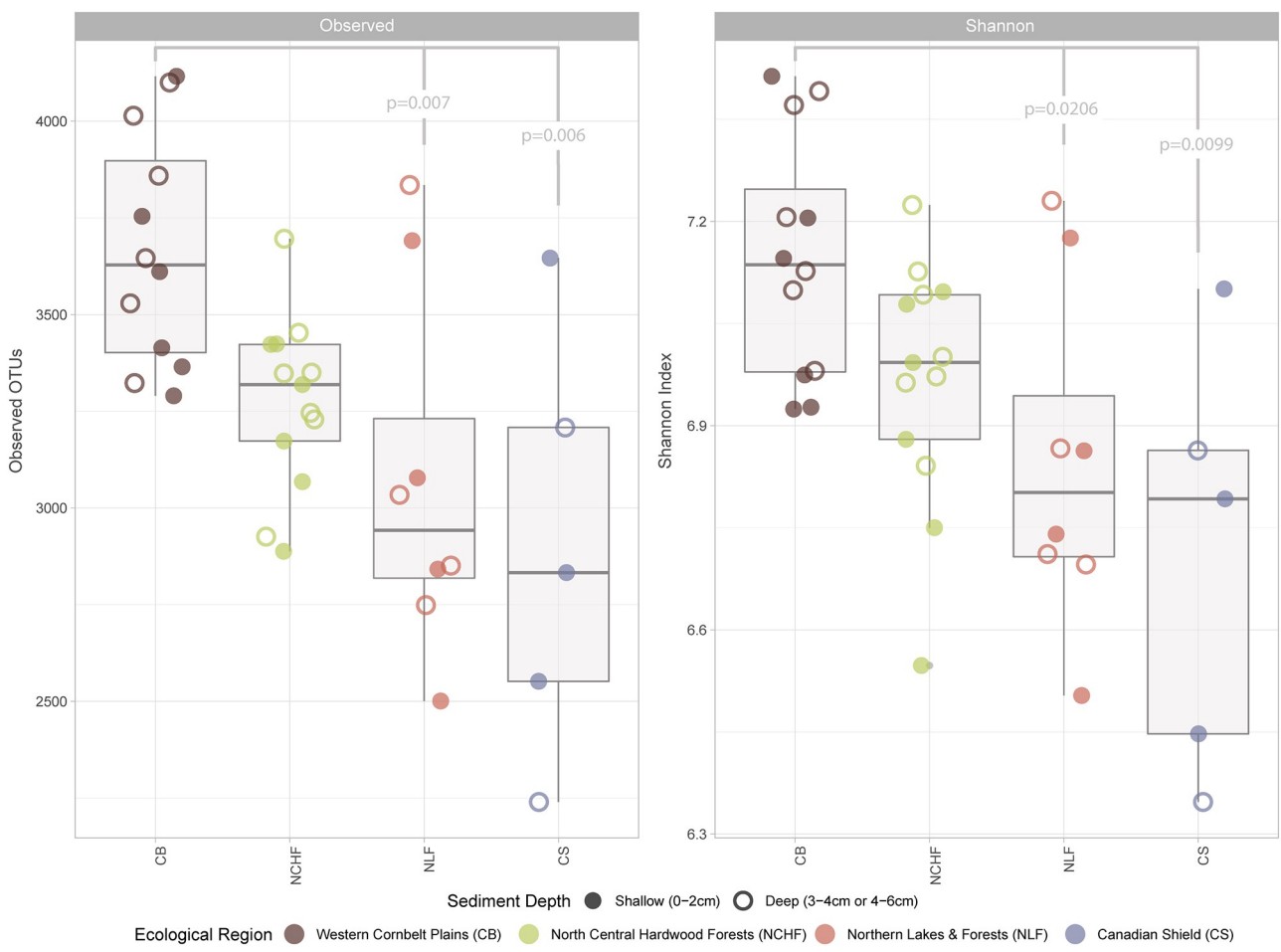

**Fig 2. Bacterial alpha diversity by ecological region.** Box plots show mean alpha level diversity of the observed Operational Taxonomic Units (OTUs) and Shannon indices of the four distinct ecological regions present within the study area: Western Cornbelt Plains (CB), North Central Hardwood Forests (NCHF), Northern Lakes and Forests (NLF), and Canadian Shield (CS). Open (shallow) and closed (deep) circles indicate unique samples and color indicates the ecological region. One sample was removed from both plots for due to uncharacteristically low diversity. Significance between regions was calculated nonparametrically using a Kruskal Wallis H test followed by a Dunn post hoc test with a Bonferroni correction. Reported p values indicate significant differences in Observed and Shannon diversity (respectively) across ecological regions, specifically the diversity of CB lake sediments when compared to NLF (p = 0.007 & p = 0.0206) and CS (p = 0.006 & p = 0.0099) sediments.

differences in the diversity levels of the CB and NLF (Dunn's test p = 0.007) and CS (Dunn's test p = 0.006) wherein the CB samples had greater richness.

The lakes in the CB are highly impacted by agricultural activity and have cultivated or pastureland comprising approximately 50–90% of their total watershed. This land use contrasts with other ecological regions like the NLF (2–24% agriculture) and the CS (<3% agriculture) where species richness was statistically less rich. Agricultural runoff and drainage can carry a variety of contaminants (e.g., herbicides, pesticides) which can stimulate the growth of certain bacteria (e.g., *Planktothrix*) [7, 51]. The selecting pressure of land-use on microbial taxa and food webs varies within and across ecosystems from highly selective to uninformative, and often land use is described as having indirect effects on microbial composition and diversity (i.e., differing land covers lead to differing nutrient loads in runoff) [22, 52, 53]. Ultimately, our data confirm patterns observed by others: that eco-regional or eco-zone concepts can affect alpha diversity and species richness [53]. We explored the eco-regional-diversity

relationship further by examining within phyla richness across the four ecoregions (S5 Table). From this we found 15 phyla which were statistically different in terms of richness in one ecoregion. Several of these phyla (e.g., Bacteroidetes, Spirochaetes and Lentisphaerae) exhibited patterns of richness across ecological regions like that seen in the entire microbiome (i.e., a decreasing richness from CB to CS) while only one phylum, Armatimonadetes, showed an opposite richness pattern. Other phyla like Epsilonbacteraeota and Modulibacteria were more diverse in the CB, while Kiritimatiellaeota were more diverse in CS samples. The identification of specific phyla that display varying richness depending on the broader ecological region of their environment may provide insight into the more nuanced indirect effects that geography plays in microbial assembly. For example, a shift in Bacteroidetes richness across these ecological regions may be indicative of the changing land use and subsequent nutrient regimes which could lead to increased algal biomass within the lake (as discussed below). However, given the low resolution of 16S rRNA gene sequencing (particularly partial gene amplicons) and an inability to confidently determine unique species and potential functional differences we could not specifically address the mechanisms that lead to increased richness across ecological regions.

Because lakes in these ecological regions also tend to vary based on their trophic status, we compare the differences in alpha diversity based on proxies for lake productivity. Using previously reported (yearly average values) of chlorophyll-a and total phosphorous (Chl-a, TP), and Secchi depth provided by the MN Department of Natural Resources, we classified the lakes as hypereutrophic, eutrophic, mesotrophic, and oligotrophic (S1 Table). We found that both Shannon diversity and observed OTUs were greater in hypereutrophic systems compared to oligotrophic systems, and eutrophic systems were also statistically richer than oligotrophic systems (S6 Fig). Since most of the lakes were classified as eutrophic or mesotrophic and the values of TP and Chl-a vary seasonally depending on sampling times, we further assessed the effects of local water chemistry, measured at time of sampling, on alpha diversity. Using an exhaustive search with AIC selection criteria we modeled Shannon and Observed diversity using all aqueous chemistry measures, lake latitude, depth, trophic status, ecoregion, and land cover use in the watershed. The lake's latitude, temperature, and specific conductivity as well as the concentrations of total phosphorus (TP), total nitrogen (TN), Chlorophyll-a best predicted Shannon diversity (adj $R^2$ = 0.586). Whereas the lake's latitude as well as the concentrations of TP, TN, DOC, specific conductivity, pH, turbidity and ecological region best predicted observed diversity (adj $R^2$ = 0.7229).

The results of our alpha diversity analysis indicate that sediments in more eutrophic systems, like those of the CB, are more diverse. This is in contrast to unimodal diversity-productivity relationships seen among other freshwater communities such as phytoplankton, zooplankton, and fish but similar to other studies on freshwater bacterial communities along a trophic gradient [54–58]. Previous work addressing the diversity-productivity relationship of bacterial communities highlights the importance of rare or dormant taxa, in that as trophic status increases the diversity of rare/dormant taxa increases [59]. In our study we deemed rare taxa at the phylum level as those not comprising more than 1% relative abundance of the sample. We then compared the richness of these phyla individually across ecological region and trophic status (S4 Table). From this, we found that no rare taxa (at the phylum level) exhibited a statistically significant (Kruskal Wallis p<0.01) increase in richness due to increased trophic status. Nevertheless, three common phyla (Chloroflexi, Spirochaetes, and TA06) did increase as a function of trophic status. Among these three phyla only Chloroflexi, which play an important role in the degradation of labile carbon and secretion of organic acids in subsurface sediments, have previously been shown to increase in abundance and diversity with eutrophication in aquatic environments [60, 61]. By examining the trends in richness across trophic

status, like ecological region, we may begin to uncover indicator taxa for nutrient pollution and eutrophication; however, coarser species and function relationships need to be considered.

## Beta diversity

To further distinguish trends in the data based on ecoregion and other environmental measures, we examined beta diversity (diversity between samples). Using our variance stabilized data, we conducted a principal component analysis (PCA) to explore the differences in community composition of samples across sites and sediment depths. The first two components explained ~30% of the variation in the samples (Fig 3). In addition to PCA we performed a cluster analysis to determine which samples were most similar (Fig 4). Using both approaches, we concluded that there was a clear distinction between community composition based on ecological region and lake depth.

In terms of ecological region, samples from the CS were the first to cluster out. The second cluster consists of six samples from three lakes that are best characterized as deep (max depth

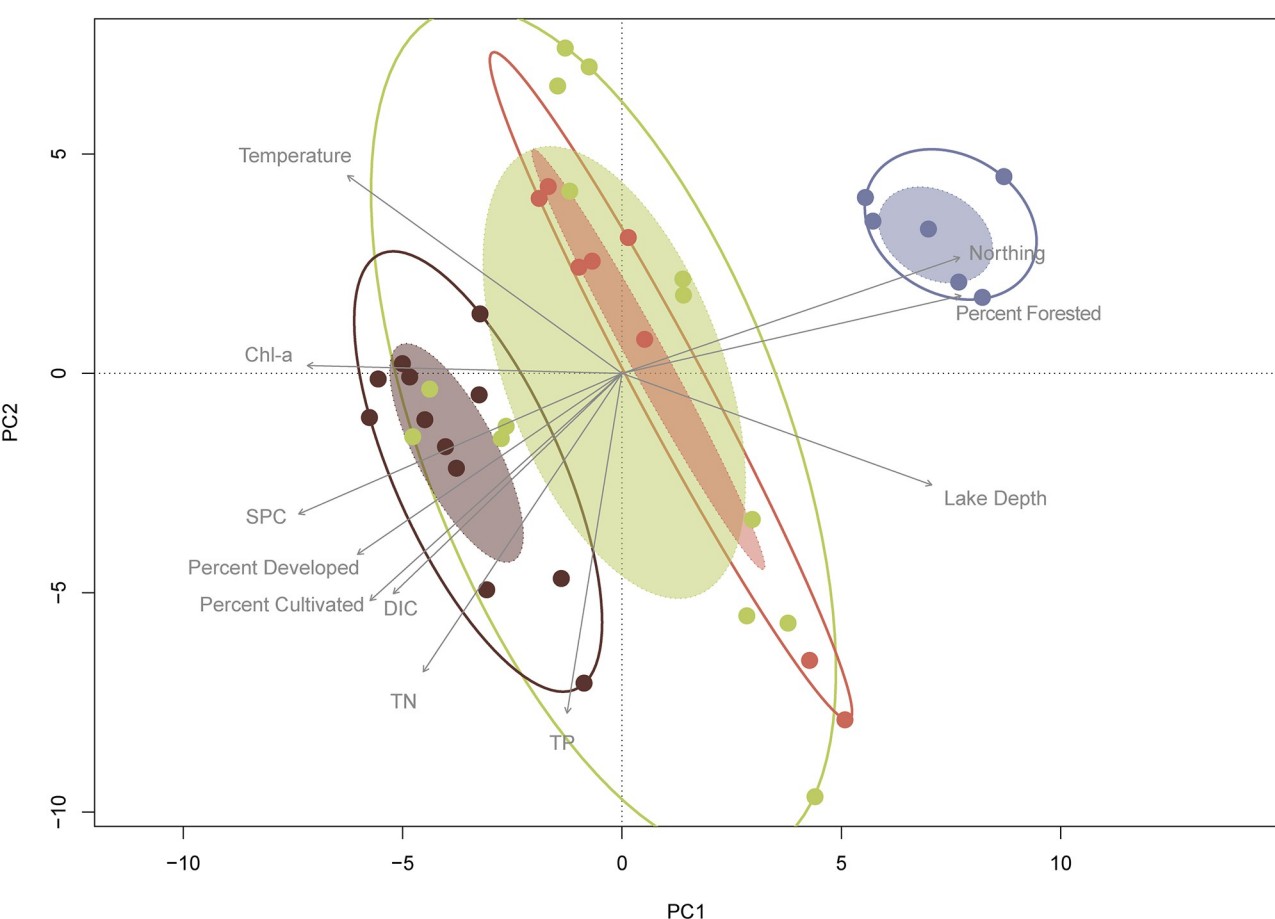

**Fig 3. Beta-diversity: Principal component analysis of samples.** Principal component analysis (PCA) of surface sediment microbiome samples where color represents ecological region. Components one and two explained 31.4% of the total variance. Environmental variables were fit using linear regressions where each component was plotted as a function of an environmental vector and those with p<0.01 were plotted. Solid line ellipses are the outer sample bounds for each region and the shaded ellipses are the standard error of the weighted centroids for the data. Abbreviations: Dissolved Inorganic Carbon (DIC), Total Nitrogen (TN), Total Phosphorus (TP), Chlorophyll-a (Chl-a), Specific Conductance (SPC).

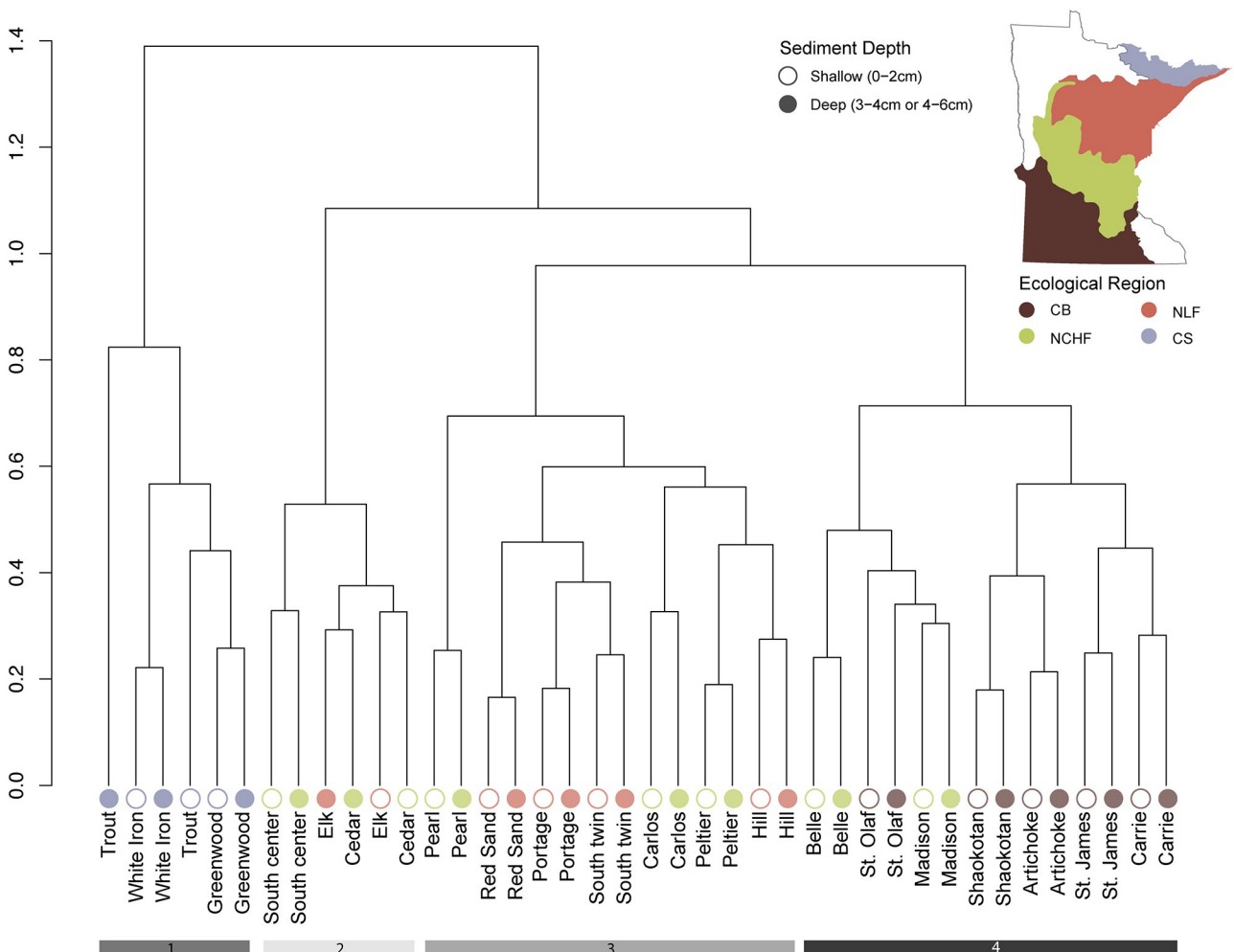

**Fig 4. Beta-diversity: Hierarchical clustering of samples.** Hierarchical clustering analysis of sediment bacterial communities using Ward's D linkages. Clusters reflect the dissimilarities (Bray-Curtis) between variance stabilized 16S rRNA OTUs within each sediment sample where shape indicates depth of samples and color ecological region. Bars along the bottom highlight the first four clusters. These clusters highlight differences in ecological region and depth, where bars 1 & 4 respectively separate Canadian Shield (CS) and Western Cornbelt Plains (CB) samples and bar 2 clusters samples from lakes ~20m or deeper. Bar 3 represents the remaining samples from the Northern Lakes and Forests (NLF) and Northcentral Hardwood Forest (NCHF) regions. Map was created using QGIS and data were made available by the MN Geospatial Commons (public domain) https://gisdata.mn.gov/.

> ~20m) meso-eutrophic systems. After depth, the CB samples cluster and the remaining cluster consists of samples from both the NLF and NCHF. Because the most predominant clusters were based on ecological region, we used a non-parametric multivariate analysis of variance (PERMANOVA) with the four regions as our independent variable and tested for differences in community dissimilarities at the OTU level using the same Bray-Curtis dissimilarity matrix as in the cluster analysis. The results of the PERMANOVA indicate there was an ecoregional difference in composition. To ensure these results reflected distinct groups and not over dispersion within groups, we used vegan's "betadisp" function to test for homogeneity of variance. The resulting insignificant values led us to conclude that community composition is a function of ecoregion and sample sites within these regions are not over dispersed.

Because ecoregion was only a consistent explanatory variable for the more geographically distant ecological regions (CB, CS), we wanted to determine if any of the local physico-chemical or watershed land-use factors were also driving community composition. We passively fit these factors to the PCA, treating them as dependent variables explained by the scores from the ordination. Each variable was analyzed separately and added to the plot where the direction of the arrow indicates the gradient direction, and the length indicates the strength of the correlation. After correcting the p-values for multiple comparisons (i.e., Bonferroni), the concentrations for DIC, TP, TN, and Chl-a and lake depth, temperature, specific conductance, and latitude were the most significant local physicochemical factors differentiating the samples. The four land uses include pastured, cultivated, developed, and forested of which only developed, forested, and cultivated lands were significant sorting factors. Forested land use was correlated in the direction of the CS whereas cultivated and urban land-uses were correlated in the opposite direction. Beyond land use, the concentration of DIC and SPC was negatively correlated with samples from lakes in the CS. Finally, the cluster of deep lakes was best explained by a combination of lake depth, temperature, and TP. These are likely related, as the greater depths of these lakes can lead to stronger and prolonged periods of thermal stratification wherein temperatures are around 4˚C and redox conditions of the sediment change to release phosphorus bound to reducible forms of iron.

The results from our study indicate that beta diversity among lake sediment microbiomes is determined by a combination of land use and productivity. These results are consistent with previous studies examining inter-lake microbiome variability across a variety of spatial scales [22, 26, 53, 58, 62]. More specifically, our work parallels findings that nitrogen is a selective variable for community composition and that it covaries with urban and agricultural land use [52]. Beyond nutrients, lake depth is commonly identified as a partitioning factor for microbial communities, as it was in our system [53]. DOC concentrations have also been strongly associated with beta diversity of microbial communities; however, our data do not reflect this trend, potentially highlighting the importance of other micronutrients and abiotic factors for explaining community variation from lake to lake [63]. Importantly, while our findings suggest a combination of productivity measures and land use are drivers of lake sediment microbiome structure, the abiotic measures used to assess these relationships were taken from water column measurements. While surface sediments are suitable for estimating site diversity, the specific vertical abiotic properties of sediments may provide deeper context to the microbiome as a whole [20, 26, 64].

## Community composition

From the 40 samples we recovered 55 unique phyla—22 of which were dominant (~90% of the total sample relative abundance) (S7 and S8 Figs). In all samples Proteobacteria were the most abundant, comprising approximately 5–20% of the total population of a given sample. Proteobacteria are often the most abundant phylum in sediment and soil ecosystems given their diverse metabolisms and role in the degradation of organic matter [65–67]. At the class level, samples were predominated largely by Deltaproteobacteria and Gammaproteobacteria and there was no clear pattern in their distribution across ecological or trophic gradients. One exception was the presence and/or high abundance of the order MBNT15 (class Deltaproteobacteria) in CS sediments (Fig 5). While there is little known about the ecological significance of MBNT15, these organisms are obligate anaerobes commonly found in stable sediments and known to reduce nitrate; as such, their presence correlates negatively with rates of nitrogen cycling [68, 69]. Additionally, they have been found to be minor constituents in sediments

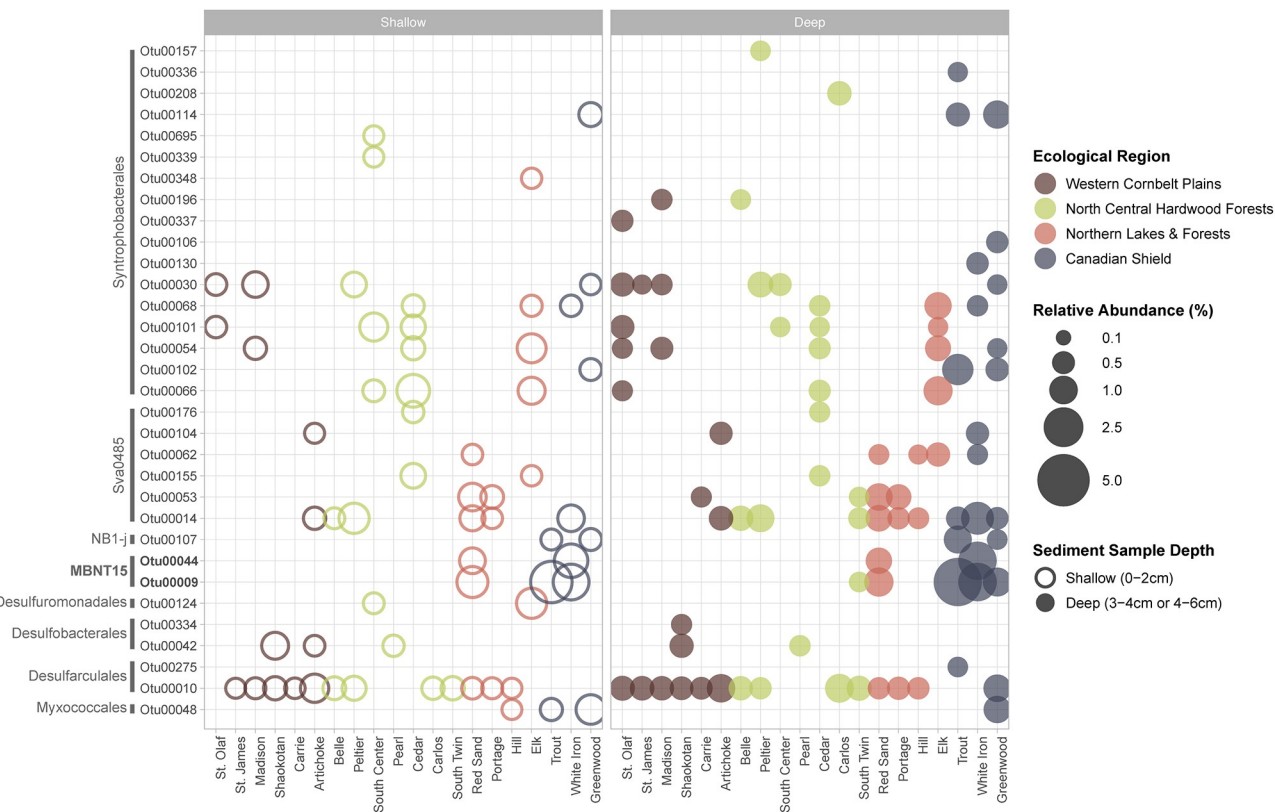

**Fig 5. Deltaproteobacteria abundance across samples.** Abundance comparison of sediment deltaproteobacterial communities where shape indicates depth of samples, color ecological region, and size the relative abundance in percent. Bars along the left group the OTUs by order. OTUs were selected if they comprised >0.05% of the total relative abundance of the sample. The OTUs bolded are mentioned in the text.

overburdened with organic matter [70]. Given the physicochemical properties of lakes in the CS region (low DOC, low DOM, and TN) our observations of greater MBNT15 abundance in the CS are not unexpected. Additionally, the presence of orders Syntrophobacterales and Desulfobacterales in several of our shallow (0-2cm) samples indicate active sulfur cycling and the depletion of oxygen in the surface sediment, as the genera within these orders are strict anaerobes [71].

Beyond Proteobacteria, other phyla like Acidobacteria, Actinobacteria, Bacteroidetes, Latescibacteria, Nitrospirae, and Spirochaetes exhibit shifts in abundance based on ecological region (S5 Fig). Latescibacteria, for example, show a subtle increase in abundance moving from the CB to the CS across the NLF and NCHF. The presence of Latescibacteria in all systems is likely due to their ability to degrade several different polymers (proteins, lipids, polysaccharides, fatty acids) and their presence in freshwater sediments supports their proposed role in algal detritus turnover [72]. The increase in abundance as a function of ecoregion may be related to the number of stratified lakes within the CS region, especially given the increase in abundance with sediment depth for most lakes.

Other phyla like Acidobacteria and Bacteroidetes display greater changes in abundance across ecological regions (S5 Fig). For example, Acidobacteria comprise <1% of the total population in CB sediments; however, in CS sediments they can make up as much as 6% of the total population. Acidobacteria, like Proteobacteria, are well distributed across environments

and metabolically diverse [73]. In soil systems these bacteria have been shown to partition based on regional land use. Notably, the order subgroup 6 (phylum Acidobacteria) has previously been observed more frequently in pastured regions compared to forested region; however, in our sediment samples these trends do not exist [74]. Subgroup 6 predominates the Acidobacteria population, is found throughout all geographic regions, and is more abundant in White Iron lake—which has one of the most heavily forested watersheds (83%). Bacteroidetes populations exhibit an opposite trend to Acidobacteria—decreased abundance in the CS sediments compared to the CB. Bacteroidetes species specialize in the degradation of organic polymers (e.g., *Bacteroides* in the gut microbiome) [75]. Given their proclivity for polymers over monomers, members of Bacteroidetes are often more abundant in aquatic systems during and shortly after algal blooms [76, 77]. Our data confirm the relationship between Bacteroidetes and productivity, as lakes in the CB region receive excess nutrient loads and often experience algal blooms during the open water season.

In addition to shifts in composition across ecological regions, there were certain phyla that were not observed at our filtering level (OTUs comprising >0.5% of the total abundance within a sample). Most notably were the absence of Cyanobacteria in CS lake sediments. Land use models have previously been used to partially estimate total cyanobacterial biomass and examine cyanobacterial community structure and genes related to the production of the cyanotoxin microcystin [78, 79]. Our data show similar findings to these previous works in which the populations of toxin-forming members like those found in the order Nostocales (S9 Fig) were more abundant in heavily agriculture and urban settings where they are well adapted to handle higher nutrient levels in the water [80].

## Caveats and conclusion

The data obtained during this study provide unique insight into the structure, diversity, and distribution of sediment bacteria across both a trophic and ecological region gradient. A primary caveat is the limited explanation of variance (~30%). While we were able to explain a similar amount of the variability to that observed in other studies, there is still a large amount unexplained [21, 52, 81]. This could be in part due to measurements that were not collected. For example, specific parameters like pH and redox potential at the sediment water interface within each lake would provide greater context for the sediment microbiome structure. Our analysis was also limited to abiotic factors as explanatory variables of alpha and beta diversity for bacterial communities. However, biotic interactions exert a selective force on community structure through a variety of control methods (e.g., grazing, phage infection) [82]. Moreover, the horizontal structuring of sediment communities as well as the overall food web dynamics especially given the differing productivity-diversity relationships could be considered in future studies.

In summary, we examined the lake sediment bacterial communities of 20 lakes to determine the influence of land-use and large-scale land classifications on community structure and diversity. We observed that ecological region with more agricultural land use and greater eutrophication exhibited higher diversity. Likewise, we found that toxin-forming community members were more abundant in heavily agriculture and urban settings. While the ability to connect changes in taxonomic composition using physio-chemical and geographical patterns is possible for some organisms, the limited resolution of short read 16S rRNA data prevent us from detecting specific taxa differences across ecological or trophic gradients.

Many land managers have access to land use maps, and remote sensing is improving our means of evaluating land use in poorly accessible parts of the world. Our results, along with future studies, offer opportunities to connect land use with sediment microbial structure and ultimately to understand lakes' abilities to adapt to anthropogenic changes.

## Supporting information

**S1 Fig. Batch effect on richness.** Observed richness (total number of OTUs) based on the total number of reads recovered per sample where color indicates the sequencing batch. There was a statistically significant Pearson's correlation between the number of total reads and the observed richness; $p < 0.001$ and R2 = 0.63.
(TIF)

**S2 Fig. Rarefaction curves.** Rarefaction curves for all forty samples in the dataset. Where each curve indicates a different sample and the vertical line is the sampling depth of 15,771 reads.
(TIF)

**S3 Fig. Alpha diversity measures by sample.** Alpha diversity measures across samples, where shape indicates depth of sample and color indicates ecological regions. All measures were calculated using Phyloseq and exhibit similar patterns in diversity; decreasing diversity across a northeasterly transect.
(TIF)

**S4 Fig. Bacterial alpha diversity across samples.** Observed diversity, a measure of richness, and Shannon diversity, measure of evenness, for all samples where shape indicates the sediment depth, color indicates the ecological region, and sites are ordered based on ecological region then latitude. One sample (Trout, Deep—CS), with lower diversity, was removed for visualization.
(TIF)

**S5 Fig. Bacterial alpha diversity across ecological region and depth.** Boxplots show mean alpha level diversity of the Observed Operational Taxonomic Units (OTUs) and Shannon indices for the four ecological regions within the study area: Western Cornbelt Plains (CB), North Central Hardwood Forests (NCHF), Northern Lakes and Forests (NLF), and Canadian Shield (CS). Samples are faceted by their sediment depth where Shallow is 0-2cm deep and Deep is 3–4 or 4-6cm deep. One deep, CS sample was removed from alpha diversity metric both plots for due to uncharacteristically low diversity.
(TIF)

**S6 Fig. Bacterial alpha diversity by trophic status.** Box plots show mean alpha level diversity of the observed Operational Taxonomic Units (OTUs) and Shannon indices of the four trophic status classifications within the study area: Oligotrophic, Mesotrophic, Eutrophic, and Hypereutrophic. Each point represents a given sample where shape indicates depth of sample and color indicates the trophic status. One sample with extremely low richness and diversity was removed from both plots for visualization. Significance between regions was calculated nonparametrically using a Kruskal Wallis H test followed by a Dunn post hoc test with a Bonferroni correction. Reported p-values indicate significant differences in Observed and Shannon diversity, respectively, across trophic status, specifically the diversity of Oligotrophic lake sediments when compared to Eutrophic ($p = 0.0224$) and Hypereutrophic ($p = 0.0038$ & $p = 0.013$) sediments.
(TIF)

**S7 Fig. Bacterial relative abundance.** Bar plots of phyla that comprise >1% of the total relative abundance of a given sample. Samples are sorted along the X axis by ecological region.
(TIF)

**S8 Fig. Bacterial abundance at the phylum level across ecological regions.** Box plots show mean relative abundance for the phyla across ecological region. Each point is a sample.

Abbreviations: Western Cornbelt Plains (CB), North Central Hardwood Forests (NCHF), Northern Lakes and Forests (NLF), and Canadian Shield (CS).
(TIF)

**S9 Fig. Cyanobacterial abundance across samples.** Abundance comparison of sediment cyanobacterial communities where shape indicates depth of samples, color ecological region, and size the relative abundance in percent. Bars along the left group the OTUs by order. OTUs were selected if they comprised >0.01% of the total relative abundance of the sample.
(TIF)

**S1 Table. Description of ecological regions.**
(XLSX)

**S2 Table. Aqueous chemistry data.**
(XLSX)

**S3 Table. Sediment extraction, DNA quality, and read depth data.**
(XLSX)

**S4 Table. Alpha diversity measures by sample.** Values for all alpha diversity measure calculated by sample.
(XLSX)

**S5 Table. Kruskal Wallis P-values for phyla richness across ecological regions and trophic status.** Table of Kruskal Wallis p-values for individual phyla within the alpha diversity dataset (rarefied). P-values for both the significance based on ecological region and trophic status are reported and bolded values are significant at p = 0.001. Dunn post hoc test with Bonferroni correction p-values are reported for those KW p<0.001. Phyla below the bolded line are phyla that comprise of <1% of the total relative abundance of a given sample (see S4 Fig).
(XLSX)

## Acknowledgments

The authors acknowledge the Minnesota Supercomputing Institute (MSI) at the University of Minnesota for providing resources that contributed to the research results reported within this paper. Michele Natarajan, Erin Mortenson, and Alaina Fedie of the St. Croix Watershed Research Station for their coordinated laboratory analyses.

## Author Contributions

**Conceptualization:** Hailey M. Sauer, Adam J. Heathcote.

**Data curation:** Hailey M. Sauer, Adam J. Heathcote.

**Formal analysis:** Hailey M. Sauer.

**Funding acquisition:** Adam J. Heathcote.

**Resources:** Trinity L. Hamilton, Rika E. Anderson, Charles E. Umbanhowar, Jr., Adam J. Heathcote.

**Supervision:** Trinity L. Hamilton, Rika E. Anderson, Adam J. Heathcote.

**Validation:** Hailey M. Sauer.

**Visualization:** Hailey M. Sauer.

**Writing – original draft:** Hailey M. Sauer.

**Writing – review & editing:** Trinity L. Hamilton, Rika E. Anderson, Charles E. Umbanhowar, Jr., Adam J. Heathcote.

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
