## [Decision Letter · Decision Letter 0]

2 Nov 2021

PONE-D-21-30070Diversity and distribution of sediment bacteria across an ecological and trophic gradientPLOS ONE

Dear Dr. Hamilton,

Thank you for submitting your manuscript to PLOS ONE. After careful consideration, we feel that it has merit but does not fully meet PLOS ONE’s publication criteria as it currently stands. Therefore, we invite you to submit a revised version of the manuscript that addresses the points raised during the review process. We encourage you to carefully consider the two major comments from Reviewer 1.

We look forward to receiving your revised manuscript.

Kind regards,

Clara Mendoza-Lera

Academic Editor

PLOS ONE

Journal Requirements:

[Funding for this project was provided by a grant to AJH from the Minnesota Environment and Natural Resources Trust fund as recommended by Legislative-Citizen Commission on Minnesota Resources (LCCMR). HMS and TLH were also supported by NSF grant #1948058. The authors acknowledge the Minnesota Supercomputing Institute (MSI) at the University of Minnesota for providing resources that contributed to the research results reported within this paper. Michele Natarajan, Erin Mortenson, and Alaina Fedie of the St. Croix Watershed Research Station coordinated laboratory analyses.]

 [Funding for this project was provided by a grant to AJH from the Minnesota Environment and Natural Resources Trust fund as recommended by Legislative-Citizen Commission on Minnesota Resources (LCCMR). HMS and TLH were also supported by NSF grant #1948058. The authors acknowledge the Minnesota Supercomputing Institute (MSI) at the University of Minnesota for providing resources that contributed to the research results reported within this paper. Michele Natarajan, Erin Mortenson, and Alaina Fedie of the St. Croix Watershed Research Station coordinated laboratory analyses.]

4. We note that Figures 1 and 5 in your submission contain [map/satellite] images which may be copyrighted. All PLOS content is published under the Creative Commons Attribution License (CC BY 4.0), which means that the manuscript, images, and Supporting Information files will be freely available online, and any third party is permitted to access, download, copy, distribute, and use these materials in any way, even commercially, with proper attribution. For these reasons, we cannot publish previously copyrighted maps or satellite images created using proprietary data, such as Google software (Google Maps, Street View, and Earth). For more information, see our copyright guidelines: http://journals.plos.org/plosone/s/licenses-and-copyright.

a) You may seek permission from the original copyright holder of igures 1 and 5 to publish the content specifically under the CC BY 4.0 license.  

Reviewers' comments:

Reviewer's Responses to Questions

**Comments to the Author**

1. Is the manuscript technically sound, and do the data support the conclusions?

Reviewer #1: Yes

Reviewer #2: Partly

2. Has the statistical analysis been performed appropriately and rigorously? 

Reviewer #1: Yes

Reviewer #2: Yes

3. Have the authors made all data underlying the findings in their manuscript fully available?

Reviewer #1: Yes

Reviewer #2: Yes

4. Is the manuscript presented in an intelligible fashion and written in standard English?

Reviewer #1: Yes

Reviewer #2: Yes

5. Review Comments to the Author

Reviewer #1: Please find below the comments to the manuscript “Diversity and distribution of sediment bacteria across an ecological and trophic gradient” by Hailey M. Sauer, Trinity L. Hamilton, Rika E. Anderson, Charles E. Umbanhowar Jr. and Adam J. Heathcote. The authors present in their manuscript an amplicon sequencing study of bacterial 16S rRNA genes from sediments of lakes in Minnesota. The aim of the study is to determine the influence of geography, land cover and physicochemical properties of the lakes on the bacterial sediment community and their diversity. Their findings include a clustering of samples due to ecological regions, trophy and lake depth.

In general, the manuscript is well written in all parts and easy to read and comprehend. The analyses they perform are sound. My biggest point of criticism is the potential lack of data that might better describe the distribution patterns.

Major:

1. The sequencing depth seems to be very shallow with only 3.3 million raw reads for 40 samples, especially when complex communities are expected in the sediment samples. Maybe the diversity in some lakes sediments is underestimated because rare taxa were missed. Please at least provide the read numbers for each lake and rarefaction curves.

2. As explanatory variables mainly the area type and measures for carbon, phosphorous and nitrogen were used. But some simple to determine variables are missing, such as water temperature, pH, conductivity, (oxygen concentration in the sediment), which are known to strongly effect some of the detected bacterial phyla, such as Proteobacteria. If these variables are neglected, the “true” causal effect might not be detectable, e.g. the effect attributed to the regions could simply be a temperature effect. Is there no additional information on the lakes available that could be included? Such as annual temperature and pH?

Minor:

1. Line 76: There are more studies available that should be cited which study communities and distribution patterns related to environmental factors, especially across Europe: e.g. https://sfamjournals.onlinelibrary.wiley.com/doi/full/10.1111/1462-2920.14992

2. The provided images have a low quality, but that might be due to the incorporation into the pdf

3. Line 99: “These regions can be characterized by their underlying geology, soils, vegetation, and land use.” Could you provide this information.

4. Line 120: How long were the samples stored before processing?

5. Nucleic Acid Preparation, Amplification, and Sequencing: Was the quality/integrity of the DNA controlled before library prep? Please provide the measures.

6. Nucleic Acid Preparation, Amplification, and Sequencing: Was the PCR performed by the core facility? What kind of PCR protocol was used?

7. Line 136: Please provide a reference for the used primers.

8. Methods: Please provide versions for all tools, programs are R packages used.

9. Methods: Were any of the environmental/physicochemical variables standardized or log transformed for any of the analyses, e.g. in the PERMANOVA? Do any of these factors covary?

10. Line 155: There is no Figure S1 in the supplement.

11. Line 158 onwards. It is not clear from the methods how the samples were grouped for the statistical tests, how many groups there were and which environmental parameters were used.

12. Line 163: Multiple (linear?) regression was used for prediction. Please have a look at https://onlinelibrary.wiley.com/doi/full/10.1111/mec.15872 were it is shown that linear predictors do not perform well on such data. Since I did not find a prediction in the results, maybe just the fitting of environmental data to the PCA axes is meant here. Please clarify.

13. Line 211: “> 2250 OTUs” is stated as diverse, but there are no references provided that compare it to other studies.

14. Fig. 2 and Fig. 3 are more or less redundant. Fig. 2 could be put into the Supplement. Further, it would be good to split the boxplots for deep and shallow sediment samples which would better show that there is not difference between these. Again, it would be good to know the sequencing depth per sample and saturation before drawing conclusion about the richness/evenness.

15. Line 255: It would be good to have some examples about functions from analysed species. Most result/discussion points provided later are on the level of phyla, classed etc. Did you not determine genera or species that perform specific function which could be attributed to ecological functions linked the different trophies?

16. Figure S3 does not indicate significant differences.

17. Line 275: What could be the reason for the more diverse community here? Are different metabolic processes involved?

18. Line 279: Rare taxa are mentioned here as important. Were these captured at this sequencing depth?

19. Line 353: You state that nitrogen is a selective variable also in your data, did you find more/less taxa for nitrogen cycling in these samples?

Reviewer #2: The manuscript hypothesizes that community composition of lakes sediments will appear homogeneous across ecological regions and land use at higher taxonomic levels but will vary at lower taxonomic levels.

The manuscript is written in an intelligible fashion and written in standard english.

I would suggest the following changes:

1. Line 31, replace morphological and chemical properties with Physico-chemical properties.

2. Line 35, add 'bacterial' community structure

3. I suggest to mention brief methods used in the study to reach the conclusion in the abstract as well.

4. Line 37, What is TP?

5. I could not find the knowledge gap or significance of the study in the whole manuscript. I suggest adding a significance statement both in the abstract and the introduction.

6. Line 61-62, needs restructuring

7. Line 76, add 'microbial/bacterial' community assembly.

8. Line 81-84, the sentence is too long and have a lot of jargon. please re-structure

9. Line 103, what year was the water sample collected.

10. How was the water sample collected and stored? Was there any treatment done to the water prior to any tests conducted? were these samples collected every-time the sediment samples were collected? why was the water analysis done only once? This is not clear.

11. How ofter were the samples collected. Details of the time the sediment samples were collected?

12. Line 126, how many samples were collected in total? You have said there were total 40 samples. How are they distributed?

13. Was there any data collected for sediment samples? e.g, pH, temperature, salinity etc?

14. Line 128, replace recommendations with protocol.

15. Line 128, How were the extraction carried out? Were they extracted in duplicates? was a certain number of samples repeated if not done in repeats? What were your controls? Both positive and negative. How can you determine the efficiency of your extraction? and how did you control for contaminations?

16. Line 136, references for your primers?

17. how were the sequences indexed?

18. What were your controls for PCR? Did you sequence your controls as well? Having negative controls is extremely important.

19. What is the distribution of these 40 samples?

20. Line 146, what version of SILVA database did you use?

21. How did you deal with controls? Did you remove any contaminant taxa?

22. Line 158, what do you mean by all available measures for alpha diversity? please mention names? Why did you choose to analyse all of them? I suggest you choose only one for each measure.

23. What metric did you choose to measure sample richness?

23. Line 168, denoising does not mean removing rare OTUs, replace word with 'filtered'.

24. Line 211, richness is not equal to diversity. What did you mean here?

25. Line 214, add name of the test used to all p values.

26. Line 283. These taxa are extremely tricky to analyse if you do not have controls. I would like you to mention the contaminants found in the sequences and then analyse this. Otherwise rare taxa data cannot be trusted.

27. Line 373, add 'relative' abundance.

28. I would suggest to draw conclusions from previous data. Did you get the same findings as from other studies?

29. Please revisit your hypothesis in the conclusion and explain in relevance to your findings.

30. Figure 3, add the name of metric to y-axis labels.

6. PLOS authors have the option to publish the peer review history of their article (what does this mean?). If published, this will include your full peer review and any attached files.

Reviewer #1: No

Reviewer #2: No

---

## [Author Response · Author response to Decision Letter 0]

16 Dec 2021

Responses to the Academic Editor

In order to comply with PLOS ONE’s style requirements, we have removed funding information from the acknowledgments section of the manuscript, changed citation styling to square brackets, and updated file names for all text, figures, and supplemental materials.

Please remove any funding-related text from the manuscript and let us know how you would like to update your Funding Statement.  Please include your amended statements within your cover letter; we will change the online submission form on your behalf.

We have removed our funding information from the main manuscript text and wish our Funding Statement to read as follows: 

Funding for this project was provided by a grant to AJH from the Minnesota Environment and Natural Resources Trust fund as recommended by Legislative-Citizen Commission on Minnesota Resources (LCCMR). HMS and TLH were also supported by NSF grant #1948058.

Please include your full ethics statement in the ‘Methods’ section of your manuscript file. In your statement, please include the full name of the IRB or ethics committee who approved or waived your study, as well as whether or not you obtained informed written or verbal consent. If consent was waived for your study, please include this information in your statement as well.

No IRB or ethics committee was required for the sample collection as we did not use human or animal subjects as part of this study. 

We note that Figures 1 and 5 in your submission contain [map/satellite] images which may be copyrighted. All PLOS content is published under the Creative Commons Attribution License (CC BY 4.0), which means that the manuscript, images, and Supporting Information files will be freely available online, and any third party is permitted to access, download, copy, distribute, and use these materials in any way, even commercially, with proper attribution. We require you to either (1) present written permission from the copyright holder to publish these figures specifically under the CC BY 4.0 license, or (2) remove the figures from your submission:

All the data used to compile the maps in Figures 1 and 4 (was Figure 5) were public domain. We have added sourcing information to each figure legend for the data. Lines 96-97 and 351.

—————————————————————————————————————————————

\fResponses to Reviewer #1 

Major: 1. The sequencing depth seems to be very shallow with only 3.3 million raw reads for 40 samples, especially when complex communities are expected in the sediment samples. Maybe the diversity in some lakes sediments is underestimated because rare taxa were missed. Please at least provide the read numbers for each lake and rarefaction curves.

We understand and appreciate this concern. We felt given our goals of examining broad scale changes in higher taxonomic level distributions that we had sufficient read depth and did not chose to resequence the samples. We have included more information on individual sample read depth in a supplemental table (S2 Table) and have added a supplemental figure (S2 Fig.) illustrating rarefaction curves.

 2. As explanatory variables mainly the area type and measures for carbon, phosphorous and nitrogen were used. But some simple to determine variables are missing, such as water temperature, pH, conductivity, (oxygen concentration in the sediment), which are known to strongly effect some of the detected bacterial phyla, such as Proteobacteria. If these variables are neglected, the “true” causal effect might not be detectable, e.g. the effect attributed to the regions could simply be a temperature effect. Is there no additional information on the lakes available that could be included? Such as annual temperature and pH?

We appreciate this review and have since added more water quality parameters to our analysis, including dissolved oxygen, temperature, pH, conductivity, and turbidity. However, as mentioned in our conclusions/caveats we do not have any sediment specific parameters. We agree with the reviewer that by neglecting some of these variables we may not be able to detect and "true" causal effect. Nevertheless, with the addition of the previously mentioned variables to our analysis there were changes in our alpha diversity models as well as the vector fitting of the PCA. Specifically, specific conductance and temperature were the two variables that correlated the strongest with the PCA axis scores and were informative to the alpha diversity models.

Minor: 1. Line 76: There are more studies available that should be cited which study communities and distribution patterns related to environmental factors, especially across Europe: e.g. https://sfamjournals.onlinelibrary.wiley.com/doi/full/10.1111/1462-2920.14992

We appreciate the reviewer’s suggestion to add further sources to this statement, and greatly appreciate the inclusion of a specific source. We have added an additional four citations to this section. We recognize that there are many more studies than the nine cited here but feel these best highlight the argument that we were making. Lines 81-82

 2. The provided images have a low quality, but that might be due to the incorporation into the pdf

We apologize for the low quality of the images provided. We did adhere to the journal requirements for exporting but have re-exported all figures in the hopes of correcting this issue.

 3. Line 99: “These regions can be characterized by their underlying geology, soils, vegetation, and land use.” Could you provide this information.

We appreciate this suggestion as it provides a greater context to the overall area of the study. We have included a new table for the supplement. Line 107

 4. Line 120: How long were the samples stored before processing?

We thank the reviewer for this comment and have since added storage times for both the sediment cores (up to 7 days) and frozen subsamples (up to three months). We have also included this information (the collection and extraction dates) in a supplemental table (S2 Table). Lines 134 & 140-141

 5. Nucleic Acid Preparation, Amplification, and Sequencing: Was the quality/integrity of the DNA controlled before library prep? Please provide the measures.

We appreciate the reviewer’s concerns about DNA quality/integrity and have added a new supplemental table (S2 Table) which includes Qubit fluorimeter readings for all DNA extractions. We've also added reference to the S2 Table in the section regarding DNA isolation where we previously commented on our final DNA concentrations. Lines 144-150

 6. Nucleic Acid Preparation, Amplification, and Sequencing: Was the PCR performed by the core facility? What kind of PCR protocol was used?

We send genomic DNA to the core facility and they perform all library prep and sequencing. We have added additional reference to the methods used by the University of Minnesota Genomic Center for further clarification. Line 162

 7. Line 136: Please provide a reference for the used primers.

This was an oversight on our part, and we have rectified it by including references for both primers. Lines 156-158

 8. Methods: Please provide versions for all tools, programs are R packages used.

Thanks for catching this! We've updated the text to include all version notes at the first appearance of the tool/program/package in the text. Lines 166, 168, 169, 174-175

 9. Methods: Were any of the environmental/physicochemical variables standardized or log transformed for any of the analyses, e.g. in the PERMANOVA? Do any of these factors covary?

We appreciate this question and apologize for not making this clear in the text. There were two circumstances where we were comparing our environmental data, first to the alpha diversity scores in the linear model and second to the axis scores of our PCA. In both circumstances we log transformed our right skewed data (e.g., TP, TN). However, in the case of the PERMANOVA we were looking at the categorical variable of ecological region, so there was no transformation necessary. Physio-chemical variables such as nutrient concentrations and chlorophyll a that describe enrichment and productivity will naturally covary along trophic gradients. This covariation is reflected in the directionality of the vectors in the PCA. Additionally, model selection using AIC penalizes for additional complexity and thus selects against explanatory variables whose explained variance is shared among 1 or more variables already included in the model.

 10. Line 155: There is no Figure S1 in the supplement.

We apologize for this and have made sure to include all supplemental and intext figures on the resubmission.

 11. Line 158 onwards. It is not clear from the methods how the samples were grouped for the statistical tests, how many groups there were and which environmental parameters were used.

We apologize for the lack of clarity and have added details on the number of samples per group to the methods section of the text. Lines 187-190

 12. Line 163: Multiple (linear?) regression was used for prediction. Please have a look at https://onlinelibrary.wiley.com/doi/full/10.1111/mec.15872 were it is shown that linear predictors do not perform well on such data. Since I did not find a prediction in the results, maybe just the fitting of environmental data to the PCA axes is meant here. Please clarify.

First, we'd like to sincerely thank the reviewer for including this source as it was incredibly informative. However, we believe there may be confusion with its relationship to this work. We used a multiple regression to model Shannon diversity and the Observed number of OTUs using our environmental data. In this case, we still had more samples than regressors and found this approach to fit the data. The results of these models were included with their R2 values in the alpha diversity subsection of the results. We did additionally fit environmental data to the PCA axes and those results are in the beta diversity subsection of the results. We regret for the lack of clarity in this approach and have refined the text to make this clearer. Lines 298-305

 13. Line 211: “> 2250 OTUs” is stated as diverse, but there are no references provided that compare it to other studies.

We thank the reviewer for this comment and have since clarified the language with this statement. We were indicating that with regard to other locations in the lake microbiome (e.g., the water column) and with regard to other sediment organisms (e.g., diatoms) these levels of diversity are great. However, our levels of diversity are similar to other studies looking at the bacterial microbiome of sediments. Lines 239-242

 14. Fig. 2 and Fig. 3 are more or less redundant. Fig. 2 could be put into the Supplement. Further, it would be good to split the boxplots for deep and shallow sediment samples which would better show that there is not difference between these. Again, it would be good to know the sequencing depth per sample and saturation before drawing conclusion about the richness/evenness. 

We have taken the reviewer's advice and moved Fig. 2 to the supplemental information. We have left Fig. 3 in the text as it reflects the samples per group for the statistical testing present in the results; however, we made an additional figure illustrating the separation of the metrics by depth (S5 Fig.). We also provide details regarding the sequencing depth per sample as mentioned in a previous comment/response.

 15. Line 255: It would be good to have some examples about functions from analysed species. Most result/discussion points provided later are on the level of phyla, classed etc. Did you not determine genera or species that perform specific function which could be attributed to ecological functions linked the different trophies?

We appreciate this comment but hesitate to attempt to assign function from a small portion of the 16S rRNA gene. We did initially perform a Tax4Fun analysis which was able to assign functions for around 1% of the data; however, the top hits were all related to housekeeping functions. In addition, at increasing taxonomic resolution, the confidence in OTU assignment decreased. A number of our OTUs were only classified (at high confidence) to the Order level. Due to the limitations of connecting function to 16S rRNA genes (e.g. https://microbiomejournal.biomedcentral.com/articles/10.1186/s40168-020-00815-y reports limited success of functional assignments to 16S rRNA amplicons outside the human microbiome), the inability of Tax4Fun to assign distinct function, and the lack of higher resolution taxonomy, we cannot determine specific functions or genera or species let alone their ecological functions across trophic levels.

 16. Figure S3 does not indicate significant differences.

We have added significant differences to what was figure S3 and is now figure S6.

 17. Line 275: What could be the reason for the more diverse community here? Are different metabolic processes involved?

See response to item 15 for context into differing metabolic processes/functions with regard to the dataset.

 18. Line 279: Rare taxa are mentioned here as important. Were these captured at this sequencing depth?

When we mention rare taxa, we're using our definition of rare in which "... we deemed rare taxa at the phylum level as those not comprising more than 1% relative abundance of the sample". Lines 312-315 While there's always the possibility that greater sequencing depth may have captured more taxa, when we discuss the importance of rare taxa we are doing so with this definition in mind.

 19. Line 353: You state that nitrogen is a selective variable also in your data, did you find more/less taxa for nitrogen cycling in these samples?

We appreciate this question and have elaborated in the text to the best of our ability about nitrogen cycling taxa. We discussed some nitrogen cycling taxa as members of the order MBNT15 and in our supplemental table 3 we do note that taxa in the phylum Nitrospirae statistically varies across the ecological regions. However, we did not discuss any specific Nitrospirae as there is only three classes, each of which contains only one order, and one family. Below the family level, our taxonomic resolution was uncultured, 6unclassified class members, or Nitrospira. Given the lack of taxonomic resolution we felt there was no substantial discussion to have surrounding the potential roles of nitrifying bacteria across these systems.

—————————————————————————————————————————————

Responses to Reviewer #2

 1. Line 31, replace morphological and chemical properties with Physico-chemical properties.

We have refined this terminology and the sentence now reads: "In this study, we selected twenty lakes with varying physio-chemical properties across four ecological regions of Minnesota." Lines 33-34 & 83-84

 2. Line 35, add 'bacterial' community structure

We have clarified this and the statement now reads: (ii) determine how lake location and watershed land-use impact aqueous chemistry and influence bacterial community structure. Lines 36-37 & 86-87 

 3. I suggest to mention brief methods used in the study to reach the conclusion in the abstract as well.

We appreciate this suggestion and have added a statement regarding the use of 16S rRNA amplicon data to our abstract. Line 37

 4. Line 37, What is TP?

We apologize for the addition of an undefined acronym here. We've corrected this to read total phosphorus instead of TP. Line 40

 5. I could not find the knowledge gap or significance of the study in the whole manuscript. I suggest adding a significance statement both in the abstract and the introduction.

We appreciate this suggestion and have added a statement of significance to the manuscript in the abstract/introduction and we revisit this significance in our conclusion. Lines 27-31 & 477-480

 6. Line 61-62, needs restructuring

We were unsure of what exactly to restructure with these lines; however, we do acknowledge that the statements were written passively in an otherwise active narrative. We've revised them to keep the same tense throughout the paragraph. Lines 65-67

 7. Line 76, add 'microbial/bacterial' community assembly.

We have clarified this sentence ensure the community assembly in question is specific to bacterial community assembly. Line 81

 8. Line 81-84, the sentence is too long and have a lot of jargon. please re-structure

We thank the reviewer for this suggestion; however, we feel that we have provided the details necessary in the preceding introduction and thus feel this wording accurately and best describes the specific aims of our study. We are also careful throughout to define terms, avoid jargon, and translate our results for both specialists and a broader audience.

 9. Line 103, what year was the water sample collected.

We apologize for not including this information in our tables. We've since updated supplemental table 2 and have included a statement about dates in the text.

 10. How was the water sample collected and stored? Was there any treatment done to the water prior to any tests conducted? were these samples collected every-time the sediment samples were collected? why was the water analysis done only once? This is not clear.

We apologize for a lack of clarity in the water sampling methodology. We have since rewritten this section to incorporate sample storage as well as sample hold times/temperatures for all measured parameters. All samples, including sediment core samples, were only taken at a single timepoint. We have also included sampling dates in our supplemental table which were accidentally left out in the first version. Lines 110-126

 11. How ofter were the samples collected. Details of the time the sediment samples were collected?

We thank the reviewer for this comment and have since added a supplemental table (S3 Table) with the exact dates of sediment sampling -- moving beyond the existing text which only clarifies the range of dates for sampling.

 12. Line 126, how many samples were collected in total? You have said there were total 40 samples. How are they distributed?

We apologize for the confusion surrounding the total number of samples and their distribution. To clarify we selected twenty lakes, each was cored and subsampled at two depth intervals resulting in the forty total samples. Following a previous comment from this reviewer, we have added a statement regarding methodology to the abstract as well as throughout the text. Lines 37-38, 137-140 & 177-178

 13. Was there any data collected for sediment samples? e.g, pH, temperature, salinity etc?

We appreciate this review and have acknowledged in our Caveats and Conclusions that our dataset is limited to water quality data only. Nevertheless, we have added water measures of pH, temperature, specific conductance, turbidity, and dissolved oxygen in our analysis. These measures, like our nutrient measures, are from a location of depth minus one meter. When added, to our analysis there were changes in our alpha diversity models as well as the vector fitting of the PCA. Specifically, specific conductance and temperature were the two variables that correlated the strongest with the PCA axis scores and were informative to the alpha diversity models.

 14. Line 128, replace recommendations with protocol.

We have changed this text to read: "We extracted DNA from 0.25g of wet sediment from each subsample using a PowerSoil DNA Isolation Kit (Qiagen, Inc.) following the manufacturer’s protocols." Lines 141-143

 15. Line 128, How were the extraction carried out? Were they extracted in duplicates? was a certain number of samples repeated if not done in repeats? What were your controls? Both positive and negative. How can you determine the efficiency of your extraction? and how did you control for contaminations?

We thank the reviewer for these comments and have addressed our use of negative controls in the text. Additionally, we provide reference to methodologies used by the University of Minnesota Genomic Center who performed negative controls for sequencing. Given the complexity of the samples we did not perform any positive control, as no mock community would adequately ensure that we sufficiently extracted the contained organisms of our samples. To address the final questions of the reviewer, extractions were only carried out once per sample. Lines 143-150 & 159-162

 16. Line 136, references for your primers?

This was an oversight on our part, and we have rectified it by including references for both primers. Lines 156-158

 17. how were the sequences indexed?

We send genomic DNA to the core facility and they perform all library prep and sequencing. We have added additional reference to the methods used by the University of Minnesota Genomic Center for further clarification. We have included reference to their procedure in the text. Line 162 -- 1. Gohl DM, Vangay P, Garbe J, MacLean A, Hauge A, Becker A, et al. Systematic improvement of amplicon marker gene methods for increased accuracy in microbiome studies. Nat Biotechnol. 2016 Sep;34(9):942–9.

 18. What were your controls for PCR? Did you sequence your controls as well? Having negative controls is extremely important.

We send genomic DNA to the core facility and they perform all library prep and sequencing, including controls. We have added additional reference to the methods used by the University of Minnesota Genomic Center for further clarification. Line 162 -- 1. Gohl DM, Vangay P, Garbe J, MacLean A, Hauge A, Becker A, et al. Systematic improvement of amplicon marker gene methods for increased accuracy in microbiome studies. Nat Biotechnol. 2016 Sep;34(9):942–9.

 19. What is the distribution of these 40 samples?

We clarify the distribution of the 40 samples in a previous comment, and in turn we've added (n=#) for all group statistical testing to clarify the distribution of samples across ecological regions. Lines 187-190

 20. Line 146, what version of SILVA database did you use?

We thank the reviewer for catching this and have since included all version notes for tools, programs, and packages at the first appearance in the text. Lines 166, 168, 169, 174-175

 21. How did you deal with controls? Did you remove any contaminant taxa?

We clarified our use of controls in the initial comment (#15); however, negative controls were sent for sequencing where they failed quality control by the core facility and were not sequenced. This information has been added to the methods section of the text. Lines 143-150 & 159-162

 22. Line 158, what do you mean by all available measures for alpha diversity? please mention names? Why did you choose to analyse all of them? I suggest you choose only one for each measure.

We thank the reviewer for this question and have included multiple measures of alpha diversity in the supplemental information because we are aware that researchers may exhibit a preference for a given metric. We chose to include them because under all examples the patterns in diversity are the same. However, in the main text we exclusively discuss the observed number of OTUs as a measure of species richness and Shannon index scores as a measure of species evenness.

 23. What metric did you choose to measure sample richness?

As stated in the text Line 186, we use the observed number of OTUs as a measure of sample richness. 

 23. Line 168, denoising does not mean removing rare OTUs, replace word with 'filtered'.

We have clarified this language in two locations in the text. Lines 200 & 207

 24. Line 211, richness is not equal to diversity. What did you mean here?

We appreciate the question surrounding our language with this statement and have since clarified this comment to reflect a comparison in richness and diversity especially with regards to the water column microbiome and diversity of diatoms in the sediments. Lines 239-242

 25. Line 214, add name of the test used to all p values.

We appreciate the reviewer’s suggestion and have added test names to all p-values throughout the text. 

  26. Line 283. These taxa are extremely tricky to analyse if you do not have controls. I would like you to mention the contaminants found in the sequences and then analyse this. Otherwise rare taxa data cannot be trusted.

We understand the reviewer’s concerns and we feel we have adequately addressed the issue of controls in previous comments (15, 18, & 21)

 27. Line 373, add 'relative' abundance.

We have made this change.

 28. I would suggest to draw conclusions from previous data. Did you get the same findings as from other studies?

We have included several discussions linking the findings in our data to those found in previous data. Lines 306-310 compared trends in alpha diversity, 385-387 contextualized our beta diversity analysis to several other studies, and 408-412, 445-449, and several others discussed specific taxonomic relationships with trophic and/or ecological status. These discussions and comparisons to existing studies served as the basis for our conclusions. 

 29. Please revisit your hypothesis in the conclusion and explain in relevance to your findings.

While we don't explicitly restate our initial aims, we do paraphrase the objectives of the study "we examined the lake sediment bacterial communities of 20 lakes to determine the influence of land-use and large-scale land classifications on community structure and diversity." We then proceed to discuss the patterns we found in alpha diversity and with regards to specific taxa. However, we did not include a statement regarding our findings about the drivers of community of composition, so we have added that to this section. We have also added additional language about the findings in relation to our study’s significance to the conclusion section. Lines 473-480

 30. Figure 3, add the name of metric to y-axis labels.

We have added the metrics to the y-axis in addition to the facets. Please note however, this figure is no longer figure 3 and is now figure 2.

---

## [Decision Letter · Decision Letter 1]

7 Mar 2022

Diversity and distribution of sediment bacteria across an ecological and trophic gradient

PONE-D-21-30070R1

Dear Dr. Hamilton,

We’re pleased to inform you that your manuscript has been judged scientifically suitable for publication and will be formally accepted for publication once it meets all outstanding technical requirements.

Kind regards,

Clara Mendoza-Lera

Academic Editor

PLOS ONE

Additional Editor Comments (optional):

Reviewers' comments:

Reviewer's Responses to Questions

**Comments to the Author**

1. If the authors have adequately addressed your comments raised in a previous round of review and you feel that this manuscript is now acceptable for publication, you may indicate that here to bypass the “Comments to the Author” section, enter your conflict of interest statement in the “Confidential to Editor” section, and submit your "Accept" recommendation.

Reviewer #2: All comments have been addressed

2. Is the manuscript technically sound, and do the data support the conclusions?

Reviewer #2: Yes

3. Has the statistical analysis been performed appropriately and rigorously? 

Reviewer #2: Yes

4. Have the authors made all data underlying the findings in their manuscript fully available?

Reviewer #2: Yes

5. Is the manuscript presented in an intelligible fashion and written in standard English?

Reviewer #2: Yes

6. Review Comments to the Author

Reviewer #2: (No Response)

7. PLOS authors have the option to publish the peer review history of their article (what does this mean?). If published, this will include your full peer review and any attached files.

Reviewer #2: No

---

## [Editor Report · Acceptance letter]

11 Mar 2022

PONE-D-21-30070R1 

Diversity and distribution of sediment bacteria across an ecological and trophic gradient 

Dear Dr. Hamilton:

I'm pleased to inform you that your manuscript has been deemed suitable for publication in PLOS ONE. Congratulations! Your manuscript is now with our production department. 

Kind regards, 

on behalf of

Dr. Clara Mendoza-Lera 

Academic Editor

PLOS ONE